# Dynamic contrast-enhanced magnetic resonance imaging parameters combined with diffusion-weighted imaging for discriminating malignant lesions, molecular subtypes, and pathological grades in invasive ductal carcinoma patients

**Gangming Zhu**[1]*, **Yongde Dong**[2☯], **Ruiting Zhu**[2☯], **Yuanman Tan**[2], **Xiao Liu**[1], **Juan Tao**[1], **Decheng Chen**[2]

**1** Department of radiology, Dongguan TungWah hospital, Dongguan, Guangdong, China, **2** Department of radiology, Dongguan Songshan Lake TungWah hospital, Dongguan, Guangdong, China

☯ These authors contributed equally to this work.

* 171849102@masu.edu.cn

**Data availability statement:** All relevant data are within the paper and its Supporting Information files.

**Funding:** This study was funded by the Dongguan Science and Technology of Social Development Program in the form of a grant

## Abstract

Dynamic contrast-enhanced magnetic resonance imaging (DCE-MRI) parameters or diffusion-weighted imaging (DWI) findings provide prognostic information on breast cancer. However, the accuracy of a single MRI technique is unsatisfactory. This study intended to explore the combination of DWI and DCE-MRI parameters in discriminating molecular subtypes in invasive ductal carcinoma (IDC) patients. Eighty-two IDC patients who underwent breast DWI and DCE-MRI examinations were retrospectively analyzed. Eighty-six patients with benign masses were retrieved as benign controls. The combination of ADC value, $K^{trans}$, $K_{ep}$, $V_e$, and iAUC had a good ability to discriminate IDC patients (vs. benign controls) with an area under the curve (AUC) [95% confidence interval (CI)] of 0.961 (0.935–0.987). A nomogram-based prediction model with the above combination showed a good predictive value for IDC probability. The combination of ADC value, $K^{trans}$, $K_{ep}$, and iAUC also had a certain ability to discriminate pathological grade III (vs. I or II) [AUC (95% CI): 0.698 (0.572–0.825)] in IDC patients. Notably, ADC value ($P$=0.010) and $K_{ep}$ ($P$=0.043) differed in IDC patients with different molecular subtypes. Besides, ADC value was increased ($P$<0.001), but $K^{trans}$ ($P$=0.037) and $K_{ep}$ ($P$=0.004) were decreased in IDC patients with Lumina A (vs. other molecular subtypes). The combination of ADC value, $K^{trans}$, $K_{ep}$, had an acceptable ability to discriminate Luminal A (vs. other molecular subtypes) [AUC (95% CI): 0.845 (0.748–0.941)] in IDC patients. DWI combined with DCE-MRI parameters discriminates IDC from benign masses; it also identifies Luminal A and pathological grade III in IDC patients.

(20211800904732) received by GZ. No additional external funding was received for this study. The funders had no role in study design, data collection and analysis, decision to publish, or preparation of the manuscript.

**Competing interests:** All authors declare that there are no conflcts of interest.

## Introduction

Breast cancer is the most commonly diagnosed cancer in females worldwide, with 2.3 million new cases and 0.68 million cancer-related deaths in 2020 [1]. Of note, invasive ductal carcinoma (IDC) is the predominant histological type of breast cancer, accounting for nearly 70% to 80% of all invasive breast cancers [2,3]. The molecular subtypes of IDC patients play a fundamental role in determining treatment and predicting prognosis [4,5]. In clinical practice, the most performed method for assessing molecular subtypes is biopsy-derived biomarkers [6]. However, the number of tissues obtained by fine needle aspiration biopsies is small, and this technique is often subject to sampling errors with invasive characteristics, which may lead to treatment failure [7]. Magnetic resonance imaging (MRI) techniques have received a lot of attention due to their advantages of no radiation, noninvasive, good spatial coverage, and high resolution of soft tissue [8,9]. In recent years, MRI techniques have shown great potential in discriminating molecular subtypes of breast cancer [10–12]. Nonetheless, a single MRI technique cannot accurately assess the molecular subtypes of IDC patients, which may further misdirect treatment decision-making and affect patients' prognosis [13,14].

Dynamic contrast-enhanced MRI (DCE-MRI) and diffusion-weighted imaging (DWI) are two common MRI techniques that provide essential information for the diagnosis of breast cancer [15]. DCE-MRI parameters, such as volume transfer constant ($K^{trans}$), diffusion of contrast medium from the extravascular extracellular leakage space (EES) back to the plasma ($K_{ep}$), volume of the EES ($V_e$), and initial area under the curve (iAUC) values, reflect vascular perfusion and permeability [16]; DWI parameter, apparent diffusion coefficient (ADC) value, can reflect the diffusion of water molecules [17]. According to different expression statuses of estrogen receptor (ER), progesterone receptor (PR), human epidermal growth factor receptor 2 (HER-2), and Ki-67, IDC patients could be classified into different molecular subtypes, including Luminal A, Luminal B, HER-2 enriched, and triple-negative breast cancer (TNBC) [18]. Their malignant tumor cell behaviors, abundance of angiogenesis, and vascular permeability are different [18,19]. Currently, although a few studies have preliminarily explored the combination of DWI and DCE-MRI parameters for discriminating molecular subtypes of breast cancer patients, more evidence is warranted [13,20].

The St. Gallen consensus defined the molecular subtypes of breast cancer [21]; according to different statuses of HER-2, luminal B could be further classified into Luminal B1 and B2. Currently, there are only a limited number of comparative studies involving Luminal B1 and B2, and further investigation is required to determine whether there are differences in DWI and DCE-MRI parameters between the two subtypes of lesions. Furthermore, in a few previous studies, a combined model incorporating DWI and DCE-MRI parameters has been utilized to enhance the predictive and diagnostic performance of IDC molecular subtypes. However, the influence of age on the model has not been taken into account, and research on adjusted models that considers age as an additional variable is relatively scarce, Consequently, the current study intended to explore the implication of DCE-MRI combined with DWI in discriminating molecular subtypes of IDC patients, the differences between Luminal B1 and B2, and the value of the adjusted model.

## Methodology

### Study population

A total of 82 IDC patients who underwent breast DWI and DCE-MRI examinations between 7th October 2021 and 2nd July 2024 were retrieved in this retrospective study. The inclusion criteria were: 1) first diagnosed as IDC; 2) with lump-like lesions; 3) with complete and

high-quality DWI and DCE-MRI imaging data; 4) had complete data of clinical characteristics. The exclusion criteria were: 1) with other non-IDC types of breast tumors; 2) with other malignancies; 3) received related treatment before DWI and DCE-MRI examinations. This study obtained approval from the Ethics Committee of Dongguan TungWah Hospital (No. 2021-KY-005). Besides, a total of 86 patients with benign masses were retrieved as benign controls. The informed consent was received from all study population or their families. The date when data were accessed for research purposes was 14th July 2024.

## MRI technique

Siemens 3.0T superconducting MR scanner (Skray, Siemens, Germany) with a dedicated 18-channel phased array surface coil was conducted for scanning. The patient was placed in a prone position. T2-weighted imaging (T2WI), DWI, and DCE-MRI scans were performed in a lateral axis scan. The scan parameters of T2WI were: TR = 4000 ms, TE = 64 ms, TI = 230 ms, layer thickness = 4 mm, layer spacing = 0.4 mm, FOV = 350 mm×350 mm, matrix = 269×384, and NEX of 2. The scan parameters of DWI were: b values = 50 s/mm$^2$ and 800 s/mm$^2$, NEX = 1 and 3, TR = 6710 ms, TE = 81 ms, FOV = 340 mm×204 mm, matrix = 160×160, layer thickness = 4 mm, and layer spacing = 0.4 mm. DCE-MRI used 3D Vibe technology with a pre-enhancement mask scan followed by injection of contrast agent Gd-DTPA (delayed for 28 s). A total of 35 consecutive phases were scanned and each phase lasted 14 s. The scan parameters of DCE-MRI were: layer thickness = 3 mm, FOV = 340 mm×340 mm, matrix = 205×256, flip angle = 15°, excitation time = 1. The intravenous injection of contrast agent Gd-DTPA (dose of 0.1 mmol/kg, rate of 2 ml/s) was followed by lateral axis scans. A total of 35 phases were dynamically collected, including plain scans and injection of contrast agent. After contrast agent injection, 15–20 ml of physiological saline was injected.

## Imaging data acquisition

The Siemens Syngo View workstation was used for image post-processing by a radiologist with 3 years of work experience. After the TISSUE 4D module was chosen, T1 mapping and DCE-MRI image data were inputted for motion correction. Then registered with anatomical maps, and blending was set to 50%. Then, the layer with the maximum lesion diameter was selected on the T1-weighted imaging (T1WI) enhanced image avoiding areas of bleeding, necrosis, and cystic. Whereafter, the region of interest (ROI) was delineated and the Kineti curve was reconstructed. The pre-evaluation parameters were set as follows: noise level of 20, MR protocol was set to dynamic, estimated T1 of 1000 ms, contrast molarity of 0.5 mmol/ml, relaxation of 3.9 l/mmol/s, and volume of 15.0 ml. When calculating the ROI, the Model Tofts was selected. Based on the quality of the curve fitting, the Fast/Middle/Low model was chosen for the Model AIF. Contrast interval time was set to 0.43 min. Finally, K$^{trans}$, K$_{ep}$, V$_e$, and iAUC values of the lesions were recorded. Besides, the layer with the maximum lesion diameter on the ADC image was selected by avoiding areas of bleeding, necrosis, and cystic. Then the ADC value of the lesion was recorded. Those above operations were repeated by another radiologist with 5 years of work experience, and rechecked by a third radiologist with 20 years of work experience. The final results were based on the consensus of the three radiologists. The images of a typical case (a left lesion from a 41-year-old woman with HER-2 enriched IDC) were shown in Fig 1, including DWI (Fig 1A), T1WI (Fig 1B), T2WI (Fig 1C), and T1 contrast-enhanced image layers (Fig 1D). The parameters of images were obtained, including values of ADC (Fig 1A, 0.790×10$^{-3}$ mm2/s), K$^{trans}$ (Fig 1E, 0.241±0.080 min-1), K$_{ep}$ (Fig 1F, 0.643±0.265 min-1), V$_e$ (Fig 1G, 0.414±0.131%), and iAUC (Fig 1H, 25.027±7.527).

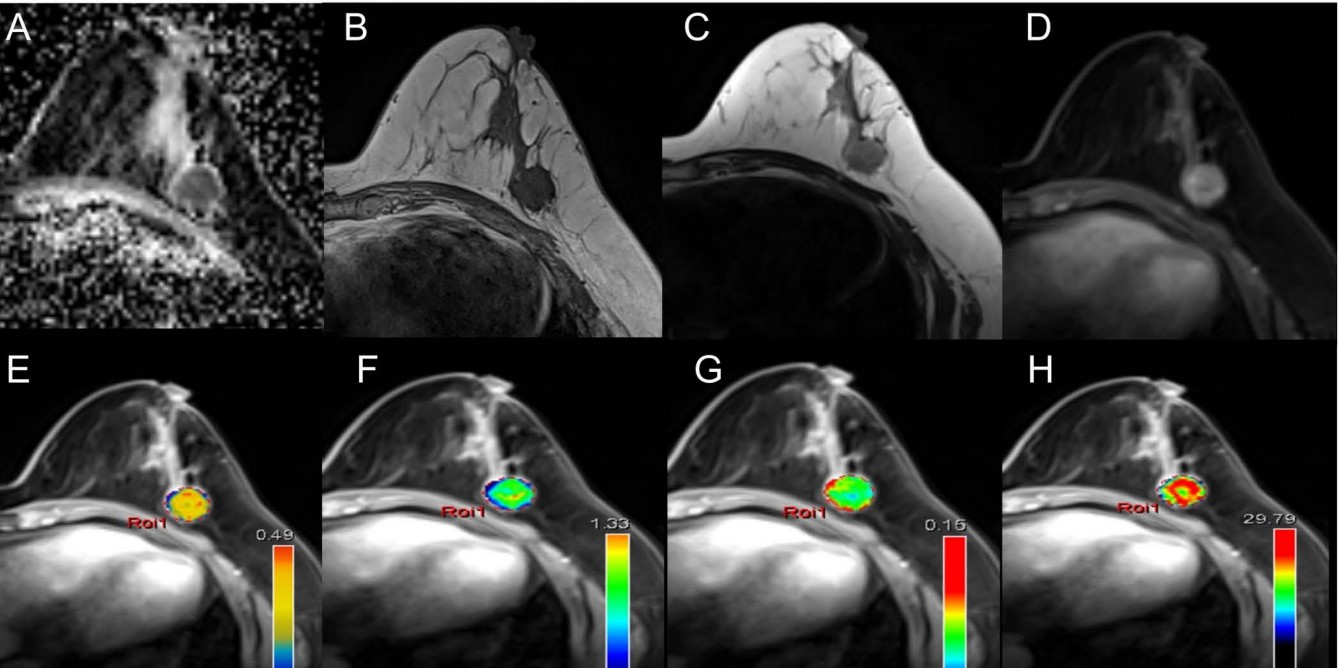

**Fig 1. MRI finding exhibition.** MRI findings of a left lesion were obtained from a 41-year-old woman with HER-2 enriched IDC; the ADC value was $0.790 \times 10^{-3}$ mm²/s (**A**); images of T1WI (**B**), T2WI (**C**), and T1 contrast-enhanced image layers (**D**); the value of $K^{trans}$ was $0.241 \pm 0.080$ min$^{-1}$ (**E**); the value of $K_{ep}$ was $0.643 \pm 0.265$ min$^{-1}$ (**F**); the value of $V_e$ was $0.414 \pm 0.131\%$ (**G**); the value of iAUC was $25.027 \pm 7.527$ (**H**).

## Clinical characteristics

Age, ER, PR, HER-2, Ki-67, molecular subtypes, and pathological grades were screened from IDC patients. The definitions of ER positive/negative, PR positive/negative, and HER-2 positive/negative were consistent with a previous study [22]. The molecular subtypes included Luminal A, Luminal B1, Luminal B2, HER-2 enriched, and TNBC. In order to reduce the sampling error caused by the small amount of tissue obtained from biopsy, this study employed core needle biopsy, and each lesion required to be punctured from multiple angles 4–8 times to obtain sufficient tissue.

## Statistical analyses

SPSS v.26.0 (IBM, USA) was applied for data analyses. The student t-test was used to compare ADC value and DCE-MRI parameters between two groups, and One-way ANOVA was used among groups. The Spearman test was used for correlation analyses. Logistics regression analyses were used for building classification models of IDC and benign control. Receiver operating characteristic (ROC) curves were displayed to show the distinguished abilities of factors or models. The performance of factors or models could be evaluated through the Youden index, which was calculated by sensitivity and specificity values. The nomogram was constructed with the combined factors (ADC value + $K^{trans}$ + $K_{ep}$ + $V_e$ + iAUC), and the calibration curve was utilized to verify model stability. A $P < 0.05$ indicated significance.

## Results

### Clinical information on IDC patients

The mean age of IDC patients was $48.3 \pm 8.8$ years. There were 54 (65.0%) patients with ER positive, 52 (63.4%) patients with PR positive, 41 (50.0%) patients with HER-2 positive, and 58

(70.7%) patients with Ki-67 positive. Regarding molecular subtypes, 16 (19.5%) patients were assessed as Luminal A, 11 (13.4%) patients were assessed as Luminal B1, 27 (32.9%) patients were assessed as Luminal B2, 14 (17.1%) patients were assessed as HER-2 enriched, and 14 (17.1%) patients were assessed as TNBC. Of note, 23 (28.1%), 33 (40.2%), and 26 (31.7%) patients were classified as pathological grade I, II, and III, respectively (Table 1).

## Discriminative ability of ADC value, DCE-MRI parameters, and their combinations between IDC patients and benign controls

ADC value was decreased in IDC patients compared to benign controls ($P<0.001$). On the contrary, $K^{trans}$ ($P<0.001$), $K_{ep}$ ($P<0.001$), and iAUC ($P<0.001$) were increased in IDC patients compared to benign controls. However, $V_e$ was not different between IDC patients and benign controls ($P=0.620$) (Table 2).

The ROC curve suggested that ADC value had a good ability to discriminate IDC patients from benign controls with an area under curve (AUC) [95% confidence interval (CI)] of 0.942 (0.905–0.979). In addition, $K^{trans}$ [AUC (95% CI): 0.826 (0.762–0.890)], $K_{ep}$ [AUC (95% CI): 0.837 (0.774–0.899)], and iAUC [AUC (95% CI): 0.748 (0.671–0.825)] had an acceptable ability to discriminate IDC patients from benign controls, but $V_e$ could not discriminate these 2 types of subjects [AUC (95% CI): 0.503 (0.415–0.591)] (Fig 2).

Surprisingly, combined model 1 [AUC (95% CI): 0.961 (0.935–0.987)], 2 [AUC (95% CI): 0.960 (0.934–0.986)], 3 [AUC (95% CI): 0.958 (0.930–0.987)], and 4 [AUC (95% CI): 0.960

**Table 1. Clinical characteristics of IDC patients.**

| Characteristics | IDC patients (N=82) |
|---|---|
| Age (years), mean±SD | 48.3±8.8 |
| ER, number (%) | |
| Positive | 54 (65.9) |
| Negative | 28 (34.1) |
| PR, number (%) | |
| Positive | 52 (63.4) |
| Negative | 30 (36.6) |
| HER-2, number (%) | |
| Positive | 41 (50.0) |
| Negative | 41 (50.0) |
| Ki-67, number (%) | |
| Positive | 58 (70.7) |
| Negative | 24 (29.3) |
| Molecular subtypes, number (%) | |
| Luminal A | 16 (19.5) |
| Luminal B1 | 11 (13.4) |
| Luminal B2 | 27 (32.9) |
| HER-2 enriched | 14 (17.1) |
| TNBC | 14 (17.1) |
| Pathological grade, number (%) | |
| Grade I | 23 (28.1) |
| Grade II | 33 (40.2) |
| Grade III | 26 (31.7) |

IDC, invasive ductal carcinoma; SD, standard deviation; ER, estrogen receptor; PR, progesterone receptor; HER-2, human epidermal growth factor receptor-2; TNBC, triple-negative breast cancer.

**Table 2. Comparison of ADC and DCE-MRI parameters between IDC patients and benign controls.**

| Items | IDC patients (N=82) | Benign controls (N=86) | P value |
|---|---|---|---|
| ADC value ($\times 10^{-3}$ mm²/s) | 0.863±0.167 | 1.517±0.397 | <0.001 |
| $K^{trans}$ (min⁻¹) | 0.944±0.715 | 0.318±0.394 | <0.001 |
| $K_{ep}$ (min⁻¹) | 1.510±1.180 | 0.551±0.754 | <0.001 |
| $V_e$ (%) | 0.673±0.211 | 0.650±0.251 | 0.620 |
| iAUC | 26.730±13.818 | 15.459±13.861 | <0.001 |

ADC, apparent diffusion coefficient; DCE-MRI, dynamic contrast-enhanced magnetic resonance imaging; IDC, invasive ductal carcinoma; $K^{trans}$, volume transfer constant from extravascular leakage space to interstitium; $K_{ep}$, rate constant from the interstitium to extravascular leakage space; $V_e$, the extracellular volume fraction; iAUC, initial area under the concentration curve.

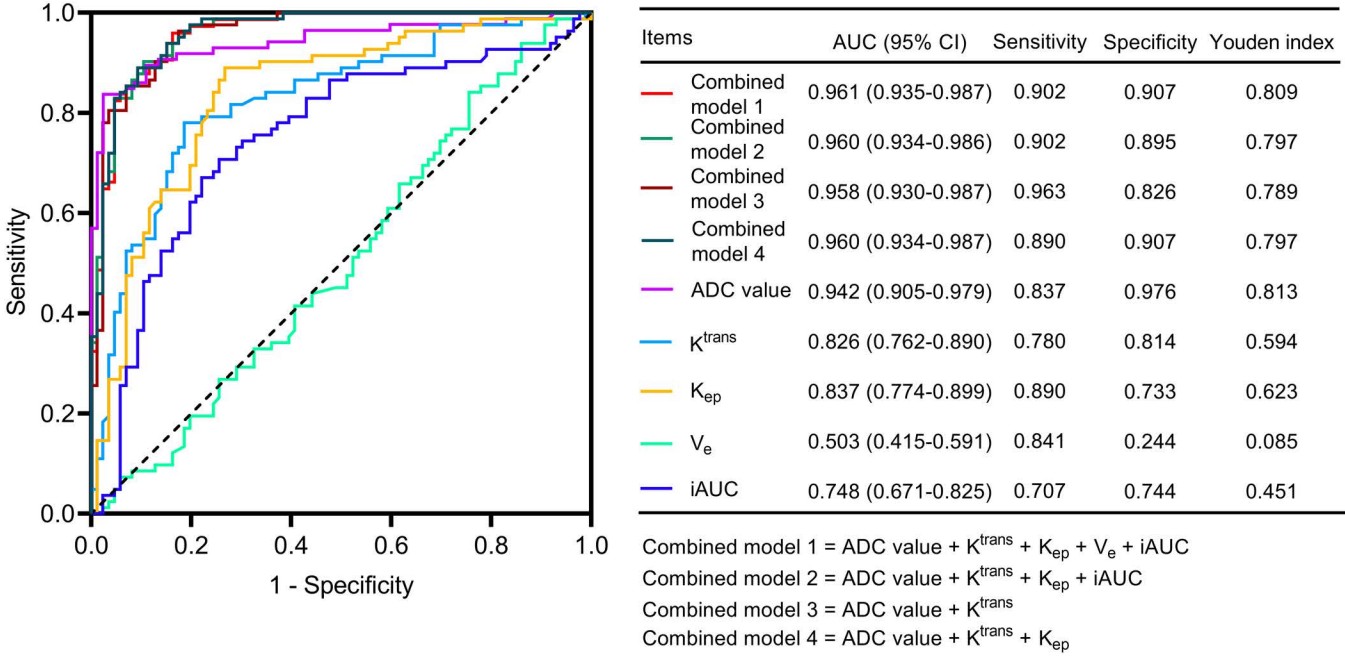

**Fig 2. Discriminative ability of DWI and DCE-MRI parameters between IDC patients and benign controls.**

(0.934–0.987)] had an excellent capacity to discriminate IDC patients from benign controls (Fig 2).

## Independent factors for distinguishing IDC patients from benign controls

According to univariable logistics regression analysis, higher ADC value was related to a lower probability of IDC versus (vs.) benign masses (*P*<0.001). higher values of $K^{trans}$ (*P*<0.001), $K_{ep}$ (*P*<0.001), and iAUC (*P*<0.001) were correlated with a higher probability of IDC vs. benign masses. Further multivariable logistics regression analysis suggested that ADC value was independently associated with a lower probability of IDC vs. benign masses [odds ratio (OR)<0.001, *P*<0.001]. On the contrary, $K^{trans}$ was independently related to a higher probability of IDC vs. benign masses (OR=7.056, *P*=0.024) (Table 3). Subsequently, a prediction model combining ADC value and DCE-MRI parameters was established through a nomogram (Fig 3A). The calibration curve of the nomogram model was exhibited in Fig 3B.

**Table 3. Logistics regression model for distinguishing IDC patients and benign controls.**

| Factors | OR | 95% CI | P value |
|---|---|---|---|
| **Univariable analysis** | | | |
| ADC value | <0.001 | <0.001-0.002 | <0.001 |
| $K^{trans}$ | 11.174 | 4.400-28.375 | <0.001 |
| $K_{ep}$ | 3.773 | 2.191-6.499 | <0.001 |
| $V_e$ | 1.396 | 0.376-5.191 | 0.618 |
| iAUC | 1.061 | 1.035-1.088 | <0.001 |
| **Multivariable analysis** | | | |
| ADC value | <0.001 | <0.001-0.005 | <0.001 |
| $K^{trans}$ | 7.056 | 1.298-38.347 | 0.024 |
| $K_{ep}$ | 1.570 | 0.608-4.052 | 0.351 |
| $V_e$ | 0.501 | 0.021-111.971 | 0.669 |
| iAUC | 0.992 | 0.944-1.042 | 0.745 |

IDC, invasive ductal carcinoma; OR, odds ratio; CI, confidence interval; ADC, apparent diffusion coefficient; $K^{trans}$, volume transfer constant from extravascular leakage space to interstitium; $K_{ep}$, rate constant from the interstitium to extravascular leakage space; $V_e$, the extracellular volume fraction; iAUC, initial area under the concentration curve.

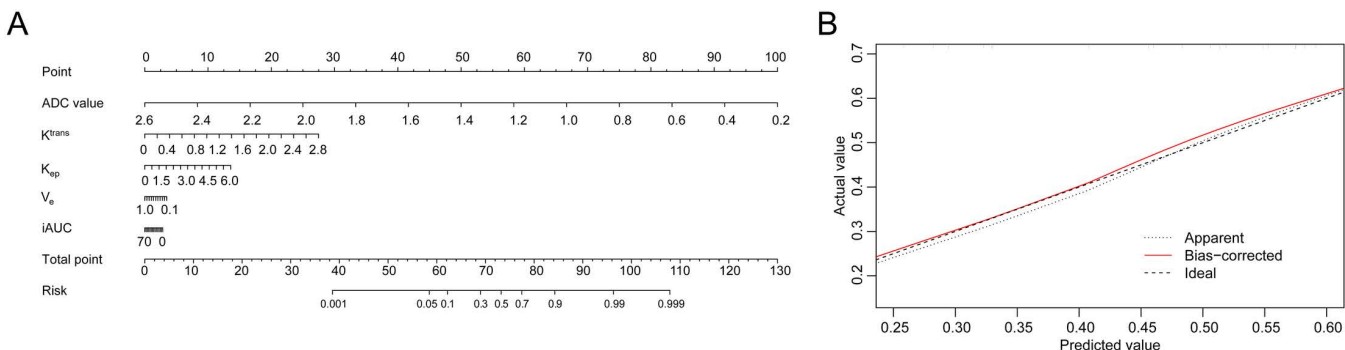

**Fig 3. Nomogram model for predicting IDC probability.** The prediction model was established through a nomogram (**A**); the calibration curve of the nomogram model (**B**).

## Relationship of ADC value and DCE-MRI parameters with ER, PR, HER-2, and Ki-67 in IDC patients

ADC value was decreased in IDC patients with Ki-67 positive vs. those with Ki-67 negative (P=0.001). At the same time, $K^{trans}$ (P=0.003) and $K_{ep}$ (P=0.007) were increased in IDC patients with Ki-67 positive vs. those with Ki-67 negative. Nevertheless, ADC value, $K^{trans}$, $K_{ep}$, $V_e$, and iAUC were not different in IDC patients with different ER, PR, and HER-2 statuses (all P>0.05) (Table 4).

## Discriminative ability of ADC value, DCE-MRI parameters, and their combinations in IDC patients with different molecular subtypes

ADC value was different among IDC patients with different molecular subtypes (P=0.010). Further comparison disclosed that ADC value was increased in IDC patients with Luminal A compared with those with other molecular subtypes (P<0.001) (Fig 4A). $K^{trans}$ did not differ among IDC patients with different molecular subtypes (P=0.181). Further comparison

**Table 4. Correlation of ADC and DCE-MRI parameters with ER, PR, HER-2, and Ki-67 of IDC patients.**

| Parameters | ADC value ($\times10^{-3}$ mm²/s) | $K^{trans}$ (min$^{-1}$) | $K_{ep}$ (min$^{-1}$) | $V_e$ (%) | iAUC |
|---|---|---|---|---|---|
| ER | | | | | |
| Positive | 0.884±0.172 | 0.838±0.676 | 1.364±1.127 | 0.669±0.212 | 25.959±14.265 |
| Negative | 0.823±0.153 | 1.147±0.757 | 1.789±1.249 | 0.679±0.213 | 28.217±13.034 |
| P value | 0.116 | 0.063 | 0.123 | 0.842 | 0.486 |
| PR | | | | | |
| Positive | 0.888±0.174 | 0.858±0.680 | 1.397±1.137 | 0.673±0.207 | 26.344±14.398 |
| Negative | 0.821±0.148 | 1.092±0.761 | 1.705±1.247 | 0.673±0.222 | 27.400±12.962 |
| P value | 0.083 | 0.154 | 0.257 | 0.993 | 0.741 |
| HER-2 | | | | | |
| Positive | 0.828±0.158 | 1.028±0.813 | 1.652±1.263 | 0.675±0.231 | 29.398±13.208 |
| Negative | 0.898±0.171 | 0.859±0.601 | 1.366±1.088 | 0.670±0.191 | 24.063±14.058 |
| P value | 0.058 | 0.288 | 0.275 | 0.915 | 0.080 |
| Ki-67 | | | | | |
| Positive | 0.826±0.140 | 1.092±0.750 | 1.733±1.202 | 0.663±0.210 | 27.100±11.821 |
| Negative | 0.953±0.196 | 0.584±0.466 | 0.968±0.943 | 0.696±0.216 | 25.836±18.026 |
| P value | 0.001 | 0.003 | 0.007 | 0.521 | 0.709 |

ADC, apparent diffusion coefficient; DCE-MRI, dynamic contrast-enhanced magnetic resonance imaging; ER, estrogen receptor; PR, progesterone receptor; HER-2, human epidermal growth factor receptor-2; IDC, invasive ductal carcinoma; $K^{trans}$, volume transfer constant from extravascular leakage space to interstitium; $K_{ep}$, rate constant from the interstitium to extravascular leakage space; $V_e$, the extracellular volume fraction; iAUC, initial area under the concentration curve.

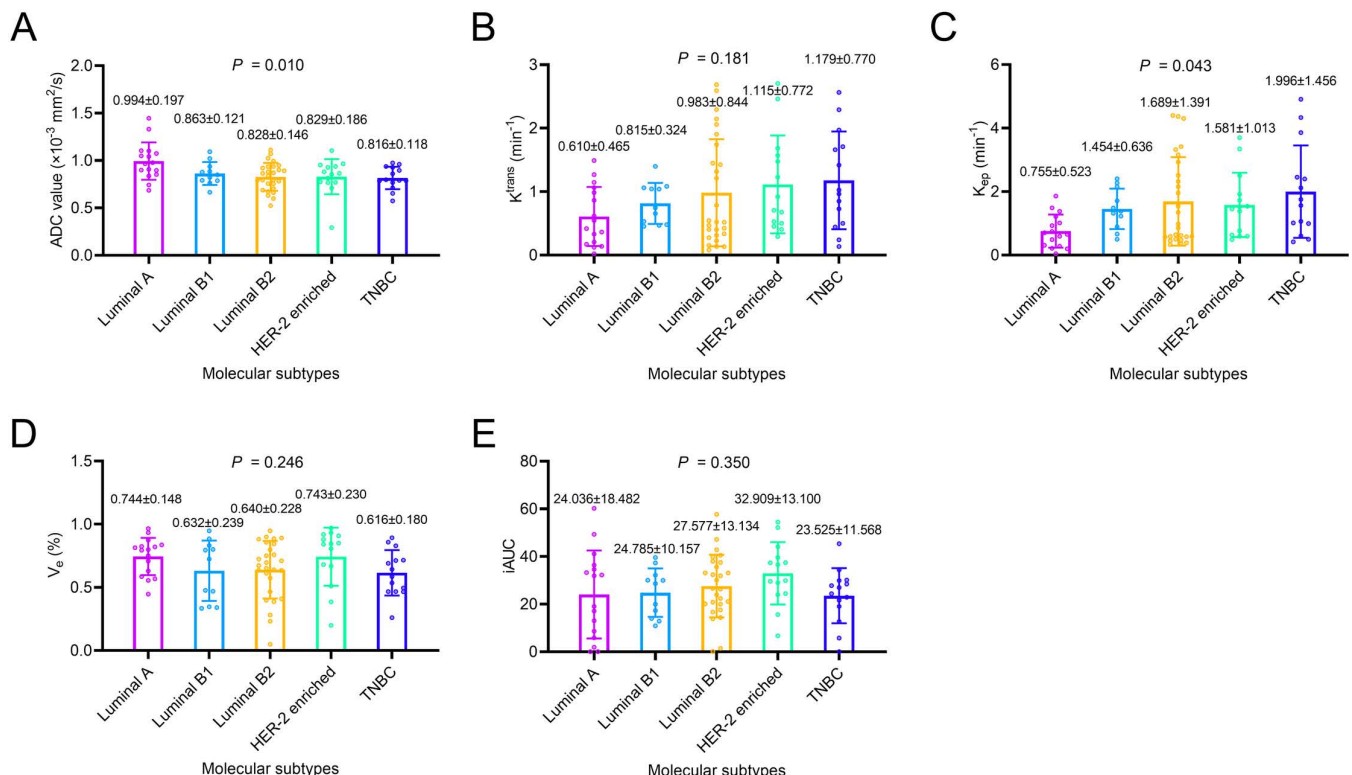

**Fig 4. Correlation of DWI and DCE-MRI parameters with molecular subtypes in IDC patients.** Relationship of ADC value (**A**), $K^{trans}$ (**B**), $K_{ep}$ (**C**), $V_e$ (**D**), and iAUC (**E**) with molecular subtypes in IDC patients.

suggested that $K^{trans}$ was decreased in IDC patients with Luminal A compared with those with other molecular subtypes ($P$=0.037) (Fig 4B). $K_{ep}$ was different among IDC patients with different molecular subtypes ($P$=0.043). Further comparison revealed that $K_{ep}$ was lower in IDC patients with Luminal A than in other molecular subtypes ($P$=0.004) (Fig 4C). $V_e$ ($P$=0.246) (Fig 4D) and iAUC ($P$=0.350) (Fig 4E) were not different among IDC patients with different molecular subtypes, and they did not differ between IDC patients with Luminal A and those with other molecular subtypes (both $P$>0.05). Of note, ADC value, $K^{trans}$, $K_{ep}$, $V_e$, and iAUC were not different between IDC patients with Luminal B1 vs. other molecular subtypes, Luminal B2 vs. other molecular subtypes, HER-2 enriched vs. other molecular subtypes, and TNBC vs. other molecular subtypes (all $P$>0.05).

According to ROC analyses, only ADC value [AUC (95% CI): 0.745 (0.601–0.890)], $K^{trans}$ [AUC (95% CI): 0.664 (0.520–0.808)], and $K_{ep}$ [AUC (95% CI): 0.743 (0.621–0.866)] had an acceptable ability to discriminate IDC patients with Luminal A from those with other molecular subtypes. Interestingly, the combined model 4 [AUC (95% CI): 0.845 (0.748–0.941)] and adjusted model 4 [AUC (95% CI): 0.850 0.746–0.955)] had a good ability to discriminate IDC patients with Luminal A from those with other molecular subtypes than other model. Besides, the combined model 4 [AUC (95% CI): 0.672 (0.512–0.832)] and adjusted model 4 [AUC (95% CI): 0.790 (0.684–0.896)] also had an acceptable capacity to discriminate IDC patients with HER-2 enriched from those with other molecular subtypes. Moreover, the combined model 1 [AUC (95% CI): 0.704 (0.578–0.830)] and adjusted model 1 [AUC (95% CI): 0.690 (0.563–0.818)] could also discriminate IDC patients with TNBC from those with other molecular subtypes (Table 5).

### Discriminative ability of ADC value, DCE-MRI parameters, and their combinations among IDC patients with different pathological grades

ADC value was different among IDC patients with grades I, II, and III ($P$=0.030). Further comparison suggested that ADC value was not different between IDC patients with grade III vs. those with grade I or II ($P$=0.134) (Fig 5A). $K^{trans}$ differed among IDC patients with grades I, II, and III ($P$=0.014). It was also increased in IDC patients with grade III vs. those with grade I or II ($P$=0.004) (Fig 5B). $K_{ep}$ was different among IDC patients with grades I, II, and III ($P$=0.032). It was elevated in IDC patients with grade III vs. those with grade I or II ($P$=0.019) (Fig 5C). $V_e$ ($P$=0.590) (Fig 5D) and iAUC ($P$=0.062) (Fig 5E) were not different among patients with grades I, II, and III, and they did not differ between IDC patients with grade III vs. those with grade I or II, either (both $P$>0.05).

ROC analyses suggested that only $K^{trans}$ [AUC (95% CI): 0.664 (0.533–0.796)] had a certain capacity to discriminate IDC patients with grade III from those with grade I or II. However, the combined model 2 [AUC (95% CI): 0.698 (0.572–0.825)] and adjusted model 2 [AUC (95% CI): 0.729 (0.601–0.856)] had an acceptable capacity to discriminate IDC patients with grade III vs. those with grade I or II (Table 6).

### Discussion

Most benign breast tumors are fibroadenomas, and the microvascular network of the benign tumors is different from breast cancer [23]. Notably, the discriminative ability of DCE-MRI and DWI between breast cancer patients and benign controls has been reported by some previous studies [24–26]. These studies report that the combination of DCE-MRI and DWI enhances the differentiating performance between malignant and benign tumors in breast cancer patients [24–26]. In the current study, it was found that ADC value was decreased, but $K^{trans}$, $K_{ep}$, and iAUC were increased in IDC patients compared to benign controls. The

**Table 5. ROC analyses of predicting different molecular subtypes of IDC patients.**

| Items | AUC | 95% CI | Sensitivity | Specificity | Youden index |
|---|---|---|---|---|---|
| **Luminal A vs. Others (Luminal B1, Luminal B2, HER-2 enriched, or TNBC)** | | | | | |
| ADC value ($\times 10^{-3}$ mm²/s) | 0.745 | 0.601-0.890 | 0.563 | 0.864 | 0.427 |
| $K^{trans}$ (min⁻¹) | 0.664 | 0.520-0.808 | 0.788 | 0.500 | 0.288 |
| $K_{ep}$ (min⁻¹) | 0.743 | 0.621-0.866 | 0.470 | 0.937 | 0.407 |
| $V_e$ (%) | 0.587 | 0.448-0.725 | 0.938 | 0.318 | 0.256 |
| iAUC | 0.559 | 0.372-0.746 | 0.773 | 0.500 | 0.273 |
| Combined model 1§ | 0.841 | 0.740-0.941 | 0.625 | 0.909 | 0.534 |
| Adjusted model 1£ | 0.847 | 0.739-0.954 | 0.813 | 0.833 | 0.646 |
| Combined model 2§ | 0.842 | 0.742-0.942 | 0.750 | 0.773 | 0.523 |
| Adjusted model 2£ | 0.852 | 0.745-0.959 | 0.813 | 0.864 | 0.677 |
| Combined model 3§ | 0.794 | 0.676-0.911 | 0.750 | 0.758 | 0.508 |
| Adjusted model 3£ | 0.801 | 0.667-0.925 | 0.813 | 0.773 | 0.586 |
| Combined model 4§ | 0.845 | 0.748-0.941 | 0.938 | 0.591 | 0.529 |
| Adjusted model 4£ | 0.850 | 0.746-0.955 | 0.813 | 0.864 | 0.677 |
| **Luminal B1 vs. Others (Luminal A, Luminal B2, HER-2 enriched, or TNBC)** | | | | | |
| ADC value ($\times 10^{-3}$ mm²/s) | 0.512 | 0.347-0.677 | 0.451 | 0.727 | 0.178 |
| $K^{trans}$ (min⁻¹) | 0.519 | 0.395-0.642 | 1.000 | 0.338 | 0.324 |
| $K_{ep}$ (min⁻¹) | 0.565 | 0.429-0.701 | 0.909 | 0.380 | 0.289 |
| $V_e$ (%) | 0.554 | 0.356-0.753 | 0.803 | 0.455 | 0.258 |
| iAUC | 0.559 | 0.397-0.721 | 0.380 | 0.818 | 0.198 |
| Combined model 1§ | 0.588 | 0.417-0.758 | 0.909 | 0.282 | 0.191 |
| Adjusted model 1£ | 0.621 | 0.466-0.776 | 0.455 | 0.831 | 0.286 |
| Combined model 2§ | 0.588 | 0.418-0.757 | 0.909 | 0.296 | 0.205 |
| Adjusted model 2£ | 0.612 | 0.468-0.756 | 0.727 | 0.606 | 0.333 |
| Combined model 3§ | 0.492 | 0.370-0.614 | 1.000 | 0.324 | 0.324 |
| Adjusted model 3£ | 0.548 | 0.430-0.666 | 1.000 | 0.423 | 0.423 |
| Combined model 4§ | 0.598 | 0.427-0.769 | 0.727 | 0.479 | 0.206 |
| Adjusted model 4£ | 0.608 | 0.469-0.748 | 0.727 | 0.606 | 0.333 |
| **Luminal B2 vs. Others (Luminal A, Luminal B1, HER-2 enriched, or TNBC)** | | | | | |
| ADC value ($\times 10^{-3}$ mm²/s) | 0.588 | 0.456-0.721 | 0.855 | 0.333 | 0.188 |
| $K^{trans}$ (min⁻¹) | 0.525 | 0.378-0.672 | 0.782 | 0.407 | 0.189 |
| $K_{ep}$ (min⁻¹) | 0.511 | 0.365-0.658 | 0.296 | 0.909 | 0.205 |
| $V_e$ (%) | 0.557 | 0.424-0.690 | 0.473 | 0.704 | 0.177 |
| iAUC | 0.534 | 0.402-0.666 | 0.926 | 0.218 | 0.144 |
| Combined model 1§ | 0.615 | 0.487-0.743 | 0.667 | 0.600 | 0.267 |
| Adjusted model 1£ | 0.624 | 0.494-0.755 | 0.630 | 0.655 | 0.285 |
| Combined model 2§ | 0.603 | 0.474-0.733 | 0.667 | 0.564 | 0.231 |
| Adjusted model 2£ | 0.593 | 0.462-0.725 | 0.630 | 0.582 | 0.212 |
| Combined model 3§ | 0.593 | 0.460-0.726 | 0.444 | 0.782 | 0.226 |
| Adjusted model 3£ | 0.583 | 0.451-0.716 | 0.259 | 0.945 | 0.204 |
| Combined model 4§ | 0.619 | 0.487-0.751 | 0.407 | 0.855 | 0.262 |
| Adjusted model 4£ | 0.615 | 0.481-0.748 | 0.407 | 0.855 | 0.262 |
| **HER-2 enriched vs. Others (Luminal A, Luminal B1, Luminal B2, or TNBC)** | | | | | |
| ADC value ($\times 10^{-3}$ mm²/s) | 0.527 | 0.370-0.685 | 0.221 | 0.929 | 0.150 |
| $K^{trans}$ (min⁻¹) | 0.596 | 0.443-0.748 | 0.929 | 0.279 | 0.208 |
| $K_{ep}$ (min⁻¹) | 0.568 | 0.422-0.715 | 0.643 | 0.588 | 0.231 |
| $V_e$ (%) | 0.651 | 0.474-0.827 | 0.571 | 0.794 | 0.365 |

*(Continued)*

**Table 5.** (Continued)

| Items | AUC | 95% CI | Sensitivity | Specificity | Youden index |
|---|---|---|---|---|---|
| iAUC | 0.653 | 0.500-0.807 | 0.857 | 0.456 | 0.313 |
| Combined model 1§ | 0.695 | 0.542-0.849 | 0.571 | 0.809 | 0.380 |
| Adjusted model 1£ | 0.790 | 0.684-0.896 | 0.786 | 0.735 | 0.521 |
| Combined model 2§ | 0.692 | 0.537-0.847 | 0.571 | 0.838 | 0.409 |
| Adjusted model 2£ | 0.793 | 0.691-0.895 | 0.857 | 0.706 | 0.563 |
| Combined model 3§ | 0.604 | 0.452-0.756 | 0.857 | 0.382 | 0.239 |
| Adjusted model 3£ | 0.764 | 0.648-0.879 | 0.857 | 0.691 | 0.548 |
| Combined model 4§ | 0.672 | 0.512-0.832 | 0.786 | 0.618 | 0.404 |
| Adjusted model 4£ | 0.790 | 0.684-0.896 | 0.786 | 0.794 | 0.580 |
| **TNBC vs. Others (Luminal A, Luminal B1, Luminal B2, or HER-2 enriched)** | | | | | |
| ADC value ($\times 10^{-3}$ mm²/s) | 0.597 | 0.447-0.747 | 0.221 | 1.000 | 0.221 |
| $K^{trans}$ (min$^{-1}$) | 0.610 | 0.447-0.773 | 0.714 | 0.544 | 0.258 |
| $K_{ep}$ (min$^{-1}$) | 0.631 | 0.470-0.793 | 0.786 | 0.485 | 0.271 |
| $V_e$ (%) | 0.613 | 0.466-0.761 | 0.529 | 0.786 | 0.315 |
| iAUC | 0.593 | 0.449-0.738 | 0.456 | 0.857 | 0.313 |
| Combined model 1§ | 0.704 | 0.578-0.830 | 0.929 | 0.515 | 0.444 |
| Adjusted model 1£ | 0.690 | 0.563-0.818 | 1.000 | 0.412 | 0.412 |
| Combined model 2§ | 0.682 | 0.548-0.815 | 0.929 | 0.426 | 0.355 |
| Adjusted model 2£ | 0.678 | 0.544-0.811 | 1.000 | 0.368 | 0.368 |
| Combined model 3§ | 0.653 | 0.502-0.805 | 0.786 | 0.500 | 0.286 |
| Adjusted model 3£ | 0.655 | 0.499-0.811 | 0.500 | 0.779 | 0.279 |
| Combined model 4§ | 0.651 | 0.507-0.795 | 0.929 | 0.397 | 0.326 |
| Adjusted model 4£ | 0.663 | 0.513-0.813 | 0.643 | 0.706 | 0.349 |

ROC, receiver operating characteristic; IDC, invasive ductal carcinoma; AUC, area under the curve; CI, confidence interval; HER-2, human epidermal growth factor receptor-2; TNBC, triple-negative breast cancer;ADC, apparent diffusion coefficient; $K^{trans}$, volume transfer constant from extravascular leakage space to interstitium; $K_{ep}$, rate constant from the interstitium to extravascular leakage space; $V_e$, the extracellular volume fraction; iAUC, initial area under the concentration curve. § Combined model 1 indicated an enter-method logistics regression model via ADC value, $K^{trans}$, $K_{ep}$, $V_e$, and iAUC. £ Adjusted model 1 indicated combined model adjusted by age.§ Combined model 2 indicated an enter-method logistics regression model via ADC value, $K^{trans}$, $K_{ep}$, and iAUC. £Adjusted model 2 indicated the combined model 2 adjusted by age. § Combined model 3 indicated an enter-method logistics regression model via ADC value and $K^{trans}$. £Adjusted model 3 indicated the combined model 3 adjusted by age. § Combined model 4 indicated an enter-method logistics regression model via ADC value, $K^{trans}$ and $K_{ep}$. £Adjusted model 4 indicated the combined model 4 adjusted by age.

potential reasons would be that: (1) ADC value reflected the diffusion of water molecules [27]. In malignant tumors, the cell growth speed was fast, and the cell density was high, which restricted the diffusion of water molecules [12,28]. Thus, ADC value was reduced in IDC patients compared to benign controls. (2) $K^{trans}$, $K_{ep}$, and iAUC represented the angiogenesis and vascular permeability [29]. Malignant tumors had abundant neovascularization and vascular perfusion, as well as high capillary permeability [30]. Therefore, $K^{trans}$, $K_{ep}$, and iAUC were increased in IDC patients vs. benign controls. Further multivariable logistic regression analysis suggested that ADC value and $K^{trans}$ could independently discriminate IDC patients from benign controls. Apart from these findings, we also proposed a nomogram model to predict the probability of IDC, and we found that the combination of DWI and MRI-DCE parameters had an excellent ability to discriminate IDC patients from benign controls, which was in line with previous studies [24,25].

The molecular subtypes of IDC possess important clinical values in treatment determination and prognosis prediction [31]. In the present study, it was found that decreased ADC value, along with $K^{trans}$ and $K_{ep}$, were associated with Ki-67 positive in IDC patients, which was

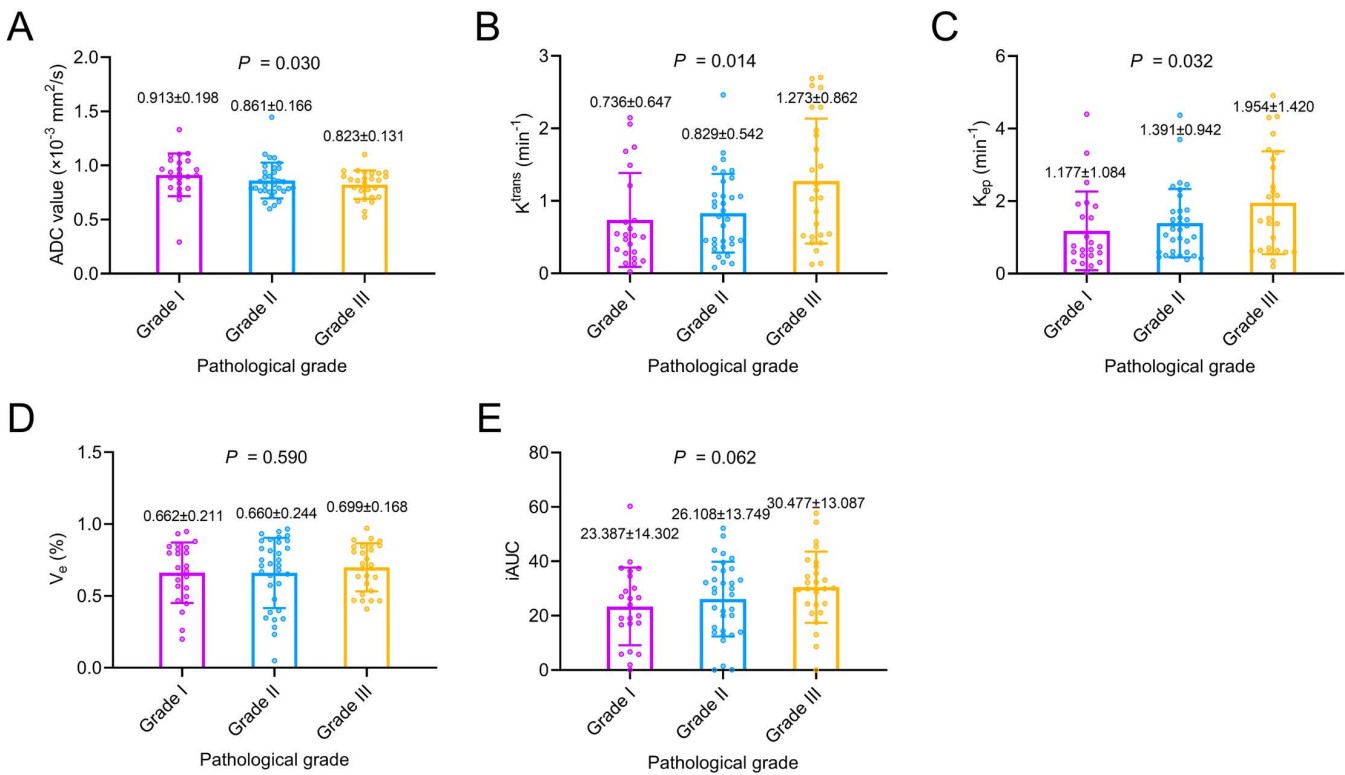

**Fig 5. Correlation of DWI and DCE-MRI parameters with pathological grades in IDC patients.** Relationship of ADC value (**A**), $K^{trans}$ (**B**), $K_{ep}$ (**C**), $V_e$ (**D**), and iAUC (**E**) with pathological grades in IDC patients.

**Table 6. ROC analyses of predicting pathological grades of IDC patients.**

| Items | AUC | 95% CI | Sensitivity | Specificity | Youden index |
|---|---|---|---|---|---|
| **Grade III vs. Grade I or II** | | | | | |
| ADC value (×10⁻³ mm²/s) | 0.595 | 0.468-0.722 | 0.321 | 0.962 | 0.283 |
| $K^{trans}$ (min⁻¹) | 0.664 | 0.533-0.796 | 0.346 | 0.929 | 0.275 |
| $K_{ep}$ (min⁻¹) | 0.628 | 0.494-0.763 | 0.462 | 0.821 | 0.283 |
| Ve (%) | 0.530 | 0.401-0.659 | 1.000 | 0.196 | 0.196 |
| iAUC | 0.614 | 0.487-0.741 | 0.846 | 0.411 | 0.257 |
| Combined model 1[§] | 0.698 | 0.572-0.825 | 0.615 | 0.696 | 0.311 |
| Adjusted model 1[£] | 0.728 | 0.600-0.856 | 0.692 | 0.750 | 0.442 |
| Combined model 2[§] | 0.698 | 0.572-0.825 | 0.654 | 0.679 | 0.333 |
| Adjusted model 2[£] | 0.729 | 0.601-0.856 | 0.692 | 0.750 | 0.442 |
| Combined model 3[§] | 0.683 | 0.554-0.812 | 0.731 | 0.571 | 0.302 |
| Adjusted model 3[£] | 0.713 | 0.581-0.845 | 0.577 | 0.821 | 0.398 |
| Combined model 4[§] | 0.685 | 0.557-0.814 | 0.731 | 0.589 | 0.320 |
| Adjusted model 4[£] | 0.715 | 0.585-0.845 | 0.692 | 0.750 | 0.442 |

ROC, receiver operating characteristic; IDC, invasive ductal carcinoma; ADC, apparent diffusion coefficient; AUC, area under the curve; CI, confidence interval; $K^{trans}$, volume transfer constant from extravascular leakage space to interstitium; $K_{ep}$, rate constant from the interstitium to extravascular leakage space; $V_e$, the extracellular volume fraction; iAUC, initial area under the concentration curve. [§] Combined model 1 indicated an enter-method logistics regression model via ADC value, $K^{trans}$, $K_{ep}$, $V_e$, and iAUC. [£] Adjusted model 1 indicated combined model adjusted by age.[§]Combined model 2 indicated an enter-method logistics regression model via ADC value, $K^{trans}$, $K_{ep}$, and iAUC. [£]Adjusted model 2 indicated the combined model 2 adjusted by age. [§]Combined model 3 indicated an enter-method logistics regression model via ADC value and $K^{trans}$. [£]Adjusted model 3 indicated the combined model 3 adjusted by age. [§]Combined model 4 indicated an enter-method logistics regression model via ADC value, $K^{trans}$ and $K_{ep}$. [£]Adjusted model 4 indicated the combined model 4 adjusted by age.

consistent with previous studies [32,33]. In addition, ADC value and $K_{ep}$ were varied among IDC patients with different molecular subtypes. ADC value was also increased in IDC patients with Luminal A compared with those with other molecular subtypes. A potential explanation would be that malignant tumor cell proliferation speed was slow in IDC patients with Luminal A vs. those with other molecular subtypes, which resulted in a small shrinkage degree of the extracellular space, leading to an increase in the diffusion of water molecules [34,35]. Therefore, ADC value was elevated in IDC patients with Luminal A compared to those with other molecular subtypes. Regarding DCE-MRI parameters, only $K^{trans}$ and $K_{ep}$ were decreased in IDC patients with Luminal A compared with those with other molecular subtypes. A reason behind this could be that angiogenesis, vascular perfusion, and vascular permeability were attenuated in IDC patients with Luminal A vs. those with other molecular subtypes, contributing to a decrease in $K^{trans}$ and $K_{ep}$ [36,37]. Hence, $K^{trans}$ and $K_{ep}$ were reduced in IDC patients with Luminal A. Surprisingly, the combined model 4 seemed to achieve an acceptable effect on discriminating different molecular subtypes of IDC patients, which was in line with previous studies [14,37]. Moreover, when we adjusted the combined model with age as an additional variable, we found that almost all models had an increase in AUC, and adjusted models had better effect on discriminating different molecular subtypes of IDC patients.

According to previous studies, DWI combined with DCE-MRI achieves a good ability to estimate pathological grades in breast cancer patients [21,38]. In the current study, it was found that the ADC value, $K^{trans}$, and $K_{ep}$ were different among IDC patients with pathological grades I, II, and III. Besides, only $K^{trans}$ and $K_{ep}$ were increased in IDC patients with pathological grade III compared to those with pathological grade I or II. We speculate the reason would be that in IDC patients with pathological grade III, the tumor was poorly differentiated, with accelerated malignant cell proliferation, as well as increased neovascularization and vascular permeability, resulting in increased $K^{trans}$ and $K_{ep}$ [22]. Therefore, higher $K^{trans}$ and $K_{ep}$ were found in IDC patients with pathological grade III compared to those with pathological grade I or II. According to ROC analysis, the combined model 2 had a good effect on discriminating the pathological grade of IDC patients. Our finding was in line with previous studies [21,38]. In addition, we also found that adjusted model 2 seems to have stronger discriminative ability than the combined model 2.

Several limitations should be mentioned in this study. (1) this was a single-center and retrospective study; thus, selection bias might unavoidably exist. (2)Although our study tried to explore the ability of DWI combined with DCE-MRI parameters to discriminate Luminal B1 and Luminal B2, the results were unsatisfactory. The small sample size might be a potential reason, and further large-scale studies were warranted to validate our findings. (3) the discriminative ability of DWI combined with DCE-MRI perfusion parameters for molecular subtypes was explored. However, the addition of MRI characterization might assist in improving their discriminative performance for molecular subtypes in IDC patients, which could be a study direction.

## Conclusions

DWI combined with DCE-MRI parameters discriminate IDC from benign masses. The combined model, particularly the adjusted models, can more effectively distinguishe Luminal A and pathological grade III in IDC patients.

## Supporting information

**S1 Data. The ADC value, DCE-MRI parameters of IDC patients and benign controls.** (XLSX)

## Author contributions

**Conceptualization:** Gangming Zhu.

**Data curation:** Gangming Zhu, YongDe Dong, Ruiting Zhu, Xiao Liu, Juan Tao, Decheng Chen.

**Formal analysis:** Yuanman Tan.

**Funding acquisition:** Gangming Zhu.

**Investigation:** Xiao Liu, Juan Tao.

**Methodology:** Gangming Zhu.

**Project administration:** Gangming Zhu, YongDe Dong, Ruiting Zhu, Yuanman Tan.

**Resources:** Ruiting Zhu.

**Software:** Decheng Chen.

**Supervision:** Gangming Zhu, Yuanman Tan.

**Writing – original draft:** Gangming Zhu, YongDe Dong, Xiao Liu, Juan Tao, Decheng Chen.

**Writing – review & editing:** Gangming Zhu, Ruiting Zhu, Yuanman Tan.

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
