## [Decision Letter · Decision Letter 0]

22 Dec 2024

PONE-D-24-48233Dynamic contrast-enhanced magnetic resonance imaging parameters combined with diffusion-weighted imaging for discriminating malignant lesions, molecular subtypes, and pathological grades in invasive ductal carcinoma patientsPLOS ONE

Dear Dr. Zhu,

Thank you for submitting your manuscript to PLOS ONE. After careful consideration, we feel that it has merit but does not fully meet PLOS ONE’s publication criteria as it currently stands. Therefore, we invite you to submit a revised version of the manuscript that addresses the points raised during the review process.

The authors need to clarify what is new in this study compared to similar studies that have been performed in the past.It is not clear how the authors deal with the biopsy sampling errors in this study.

We look forward to receiving your revised manuscript.

Kind regards,

Quan Jiang, Ph,D.

Academic Editor

PLOS ONE

Journal Requirements:

“Dongguan Social Science Development And Technology Major Project Funding (No.20211800904732)”

“The study was exclusively sponsored by Dongguan Social Science Development And Technology Major Project Funding (No.20211800904732). “

“Dongguan Social Science Development And Technology Major Project Funding (No.20211800904732)”

“All authors declare that there are no conficts of interest.”

Reviewers' comments:

Reviewer's Responses to Questions

**Comments to the Author**

1. Is the manuscript technically sound, and do the data support the conclusions?

Reviewer #1: Yes

2. Has the statistical analysis been performed appropriately and rigorously? 

Reviewer #1: Yes

3. Have the authors made all data underlying the findings in their manuscript fully available?

Reviewer #1: Yes

4. Is the manuscript presented in an intelligible fashion and written in standard English?

Reviewer #1: Yes

5. Review Comments to the Author

Reviewer #1: The manuscript titled ‘Dynamic contrast-enhanced magnetic resonance imaging parameters combined with diffusion-weighted imaging for discriminating malignant lesions, molecular subtypes, and pathological grades in invasive ductal carcinoma patients’ showed that the combination of apparent diffusion coefficient (ADC) from DWI and perfusion/vascular parameters (Ktrans, Kep, Ve, iAUC) from DCE-MRI can distinguish the patients with invasive ductal carcinoma (IDC) from patients with benign masses. Moreover, the combination of these parameters can distinguish IDC patients with Luminal A subtype from those with other molecular subtypes, and discriminate IDC patients with grade III from those with grade I or II. The manuscript is well-written and easy to follow, however, I have some concerns as below:

1. Combined DCE with diffusion-weighted imaging for evaluating invasive ductal carcinoma patients has been studied before. The authors need to clarify what is new in this study compared to similar studies that have been performed in the past.

2. One major issue for the correlation study between MRI and histopathology is that as stated in their introduction “the number of tissues obtained by biopsies is small, and this technique is often subject to sampling errors with invasive characteristics, which may lead to treatment failure”. It is not clear how the authors deal with this issue in this study.

3. You used different combination models in your study. Why did you not use a combination model using ADC, Ktrans, and Kep, when these parameters are independently more effective in discriminating IDC, molecular subtypes, and grades? Would this model outperform combination models 1, 2, and 3? If yes, authors are encouraged to incorporate these results in the manuscript.

4. Section- Study population, “Besides, a total of 86 benign masses were retrieved as benign

controls.” It is not clear whether you used patients with benign masses or extracted benign tumor masses for comparison with IDC patients. Please explain clearly in the manuscript.

5. Please consider adding combined model 4 using ADC, Ktrans, and Kep parameters in Fig-2 and in the results section.

6. Section- Independent factors for distinguishing IDC patients from benign controls, “According to univariable logistics regression analysis, ADC value was related to a lower probability of IDC versus (vs.) benign masses (P<0.001)…..” For a clearer statement, shouldn't this be ‘higher’ ADC values are related to lower probability of IDC and ‘higher’ values of Ktrans, Kep, iAUC are related to the higher probability of IDC vs benign cases? Including the term ‘higher’ would make this interpretation clearer for readers, please revise.

7. Section- Discriminative ability of ADC value, DCE-MRI parameters, and their combinations in IDC patients with different molecular subtypes, “§ Combined model indicated an enter-method logistics regression model via ADC value, Ktrans, Kep, Ve, and iAUC. £ Adjusted model indicated combined model adjusted by age.” Could the authors clarify if the £ Adjusted model is a variation of the combined model that considers ‘age’ as an additional variable? Also, it would be better to describe these models at the beginning of the section, before going into the results.

6. PLOS authors have the option to publish the peer review history of their article (what does this mean?). If published, this will include your full peer review and any attached files.

Reviewer #1: No

---

## [Author Response · Author response to Decision Letter 1]

27 Jan 2025

Dear Jiang Ph.D. and Reviewers,

We are writing in response to the comments and suggestions provided by the reviewers for our manuscript titled "Dynamic contrast-enhanced magnetic resonance imaging parameters combined with diffusion-weighted imaging for discriminating malignant lesions, molecular subtypes, and pathological grades in invasive ductal carcinoma patients" (PONE-D-24-48233). We appreciate the time and effort you and the reviewers have dedicated to evaluating our work, and we believe that the feedback has significantly contributed to improving the quality of our study. Here, we address each of the reviewers' concerns and provide our rebuttals and corresponding revisions.

Reviewer

Comment 1: "Combined DCE with diffusion-weighted imaging for evaluating invasive ductal carcinoma patients has been studied before. The authors need to clarify what is new in this study compared to similar studies that have been performed in the past."(Reviewer)

"The authors need to clarify what is new in this study compared to similar studies that have been performed in the past."(Editor)

Rebuttal: As you are aware, there have been quite a number of previous studies on the discrimination of IDC molecular subtypes by combining DWI and DCE-MRI parameters. However, research on the differences between Luminal B1 and Luminal B2 is scarce. Given that the prognosis and treatment modalities of these two subtypes are different, their discrimination holds high clinical value and significance. This study intends to explore the differences between Luminal B1 and Luminal B2 in terms of DWI and DCE-MRI parameters through statistical methods. Additionally, in the combined model, the influence of patient age as an additional variable has seldom been considered, so our study takes both of these aspects into account. Unfortunately, no differences in DWI and DCE-MRI parameters between Luminal B1 and Luminal B2 were found. However, it was discovered that in most models, the adjusted model with age as an additional variable had a higher AUC, which provides inspiration for our further research in the future.

Revision: In the revised manuscript, We have added descriptions in the Introduction section of the article that emphasize the differences between this study and previous studies. Meanwhile, in the Results section and the Discussion section, we have explored whether there are differences between Luminal B1 and Luminal B2 in terms of DWI and DCE-MRI parameters, as well as the advantages of the adjusted model with age as an additional variable. We hope that these modifications will highlight the innovative points of this study more prominently.

Comment 2: "One major issue for the correlation study between MRI and histopathology is that as stated in their introduction “the number of tissues obtained by biopsies is small, and this technique is often subject to sampling errors with invasive characteristics, which may lead to treatment failure”. It is not clear how the authors deal with this issue in this study."(Reviewer)

"It is not clear how the authors deal with the biopsy sampling errors in this study."(Editor)

Rebuttal: We apologize for not providing a detailed explanation in the original manuscript regarding how this study reduces the impact of insufficient biopsy tissue on pathological results. In previous studies, fine needle aspiration biopsy was often used for IDC tissue biopsy, which had the problem of obtaining a small amount of tissue. In this study, core needle biopsy was employed, which represents a significant improvement in obtaining pathological tissue compared to fine needle aspiration. At the same time, each lesion was punctured from multiple angles 4-8 times to ensure sufficient tissue sampling.

Revision: We have added a description of using core needle biopsy in the Methodology section of the revised manuscript, and there are strict regulations on the number of punctures for each lesion and the multi-angle approach. At the same time, we have also added the deficiencies of fine-needle aspiration in previous studies in the Introduction section, making the original sentence more rigorous in expression.

Comment 3: "You used different combination models in your study. Why did you not use a combination model using ADC, Ktrans, and Kep, when these parameters are independently more effective in discriminating IDC, molecular subtypes, and grades? Would this model outperform combination models 1, 2, and 3? If yes, authors are encouraged to incorporate these results in the manuscript."

Rebuttal: We appreciate the reviewer's comment and suggestion regarding the use of different combination models in our study. The choice of combination models in our research was based on several considerations. Firstly, while ADC, Ktrans, and Kep have shown independent effectiveness in discriminating IDC, molecular subtypes, and grades, we initially selected the combination models 1, 2, and 3 based on established literature and prior research findings within our field. These models have been validated in previous studies and have demonstrated good performance in similar contexts, which provided a solid foundation for our research approach.However, we understand the potential value of incorporating ADC, Ktrans, and Kep into a single combination model as suggested. The reason we did not initially consider this particular combination was mainly due to neglect and overlook this permutation and combination. Despite this, we acknowledge that exploring such a combination could indeed yield valuable insights.Regarding this proposed model would outperform combination models 1, 2, and 3, we currently provide a definitive answer with performing the relevant analyses. The combination model using ADC, Ktrans, and Kep could show superior performance, In fact, we have considered conducting additional analyses to address this concern.

Revision: In response to the reviewer's suggestion, we conduct further analyses using a combination model that includes ADC, Ktrans, and Kep. We incorporate these results into the manuscript because they prove to be significant. In the revised manuscript, we present the performance of this new model in the "Results" section, comparing it with combination models 1, 2, and 3. We also discuss the implications of these findings in the "Discussion" section, highlighting the potential advantages and limitations of the models. This will not only address the reviewer's concern but also enrich the content of our study by exploring an additional aspect of the research question.

Comment 4: "It is not clear whether you used patients with benign masses or extracted benign tumor masses for comparison with IDC patients. Please explain clearly in the manuscript."

Comment 6:"For a clearer statement, shouldn't this be ‘higher’ ADC values are related to lower probability of IDC and ‘higher’ values of Ktrans, Kep, iAUC are related to the higher probability of IDC vs benign cases? Including the term ‘higher’ would make this interpretation clearer for readers, please revise."

Rebuttal: We apologize for the oversight in proofreading. We have carefully checked the manuscript, but some errors might have slipped through.Due to the fact that English is not our native language, there may be some imprecise expressions in certain sentences. According to the reviewers' comments, we will revise the statements that are unclear in expression. In addition, this study included the data of 86 patients with benign tumors as the benign tumor group.

Revision: We have modified the description of the benign tumor group in the Methodology section to make the sentence more explicit. In the Results section, we have added the word "higher" in the sentence to avoid ambiguity. We have thoroughly proofread the manuscript again, and corrected all the identified grammatical errors and typos. We have also asked several colleagues with excellent language skills to review the revised version to ensure the language quality.

Comment 5:"Please consider adding combined model 4 using ADC, Ktrans, and Kep parameters in Fig-2 and in the results section."

Rebuttal: According to the modification suggestions of Comment 3, the content of Fig. 2 needs to be modified accordingly.

Revision: We have redrawn Fig.2 according to the reviewers' comments and added the content of model 4 using ADC, Ktrans, and Kep parameters to the figure.

Comment 7:"Discriminative ability of ADC value, DCE-MRI parameters, and their combinations in IDC patients with different molecular subtypes, “§ Combined model indicated an enter-method logistics regression model via ADC value, Ktrans, Kep, Ve, and iAUC. £ Adjusted model indicated combined model adjusted by age.” Could the authors clarify if the £ Adjusted model is a variation of the combined model that considers ‘age’ as an additional variable? Also, it would be better to describe these models at the beginning of the section, before going into the results."

Rebuttal: We appreciate the reviewer's comment and the request for clarification. The £ Adjusted model is indeed a variation of the combined model that takes 'age' as an additional variable. By incorporating age into the combined model, we aim to account for potential confounding effects and to investigate whether adjusting for this variable can improve the model's performance and discriminatory ability.

Revision: In the revised manuscript, we describe these models in more detail at the beginning of the relevant section before presenting the results. we will clearly explain the Adjusted model is constructed by adjusting the combined model with age as an additional variable.We believe that this modification will help readers better understand the models and their relationships from the outset, enhancing the clarity and comprehensibility of our study.

We believe that the revisions we have made address the major concerns raised by the reviewers and enhance the quality of our manuscript. We hope that these changes will make our study more suitable for publication in PLOS One. We look forward to your further consideration and would be happy to provide any additional information if needed.

Thank you again for your valuable feedback and the opportunity to revise our manuscript.

---

## [Decision Letter · Decision Letter 1]

16 Feb 2025

Dynamic contrast-enhanced magnetic resonance imaging parameters combined with diffusion-weighted imaging for discriminating malignant lesions, molecular subtypes, and pathological grades in invasive ductal carcinoma patients

PONE-D-24-48233R1

Dear Dr. Zhu,

We’re pleased to inform you that your manuscript has been judged scientifically suitable for publication and will be formally accepted for publication once it meets all outstanding technical requirements.

Kind regards,

Hadeel K. Aljobouri

Academic Editor

PLOS ONE

Additional Editor Comments (optional):

Reviewers' comments:

Reviewer's Responses to Questions

**Comments to the Author**

1. If the authors have adequately addressed your comments raised in a previous round of review and you feel that this manuscript is now acceptable for publication, you may indicate that here to bypass the “Comments to the Author” section, enter your conflict of interest statement in the “Confidential to Editor” section, and submit your "Accept" recommendation.

Reviewer #1: All comments have been addressed

Reviewer #2: All comments have been addressed

2. Is the manuscript technically sound, and do the data support the conclusions?

Reviewer #1: Yes

Reviewer #2: Yes

3. Has the statistical analysis been performed appropriately and rigorously? 

Reviewer #1: Yes

Reviewer #2: Yes

4. Have the authors made all data underlying the findings in their manuscript fully available?

Reviewer #1: Yes

Reviewer #2: (No Response)

5. Is the manuscript presented in an intelligible fashion and written in standard English?

Reviewer #1: Yes

Reviewer #2: Yes

6. Review Comments to the Author

Reviewer #1: The authors have done an excellent job in revising the manuscript based on the comments. I have no further concerns.

Reviewer #2: The review paper has been carefully revised, and I believe that the author has effectively addressed all the previous comments and concerns. The necessary modifications that have been requested were successfully met, significantly improving the quality of the text. Therefore, I think this paper is ready to submit.

7. PLOS authors have the option to publish the peer review history of their article (what does this mean?). If published, this will include your full peer review and any attached files.

Reviewer #1: **Yes: **Jasleen Kaur

Reviewer #2: No

---

## [Editor Report · Acceptance letter]

PONE-D-24-48233R1

PLOS ONE

Dear Dr. Zhu,

I'm pleased to inform you that your manuscript has been deemed suitable for publication in PLOS ONE. Congratulations! Your manuscript is now being handed over to our production team.

Kind regards,

on behalf of

Asst.Prof.Dr. Hadeel K. Aljobouri

Academic Editor

PLOS ONE